### Regional soil moisture monitoring network in the Raam catchment in the Netherlands

Harm-Jan F. Benninga<sup>1</sup>, Coleen D.U. Carranza<sup>2</sup>, Michiel Pezij<sup>3</sup>, Pim van Santen<sup>4</sup>, Martine J. van der Ploeg<sup>2</sup>, Denie C.M. Augustijn<sup>3</sup>, Rogier van der Velde<sup>1</sup>

<sup>1</sup>Water Resources, Faculty of Geo-Information Science and Earth Observation, University of Twente, 7500 AE Enschede, 5 The Netherlands <sup>2</sup>Soil Physics and Land Management Group, Department of Environmental Sciences, Wageningen University, 6700 AA Wageningen, The Netherlands <sup>3</sup>Water Engineering and Management, Faculty of Engineering Technology, University of Twente, 7500 AE Enschede, The 10 Netherlands <sup>4</sup>Waterschap Aa en Maas, 5216 PP 's-Hertogenbosch, The Netherlands

Correspondence to: Harm-Jan F. Benninga (h.f.benninga@utwente.nl)

Abstract. We have established a soil moisture profile monitoring network in the 223 km<sup>2</sup> Raam Catchment, a tributary of the Meuse River in the Netherlands. This catchment faces water shortage during summers and excess of water during winters

- and after extreme precipitation events. Water management can benefit from reliable information on the soil water availability 15 in the unsaturated zone. In situ measurements provide a direct source of information on which water managers can base their decisions. Moreover, these measurements are commonly used as a reference for calibration and validation of soil moisture products derived from earth observations or obtained by model simulations. Distributed over the Raam Catchment, we have equipped 14 agricultural fields and one natural grass field with soil moisture and soil temperature monitoring
- instrumentation, consisting of Decagon 5TM sensors installed at depths of 5 cm, 10 cm, 20 cm, 40 cm and 80 cm. Soil-20 specific calibration functions that have been developed for the 5TM sensors under laboratory conditions lead to an accuracy of 0.02 m<sup>3</sup> m<sup>-3</sup>. The first set of measurements has been retrieved for the period 5 April 2016 – 4 April 2017. In this paper, we describe the Raam monitoring network and instrumentation, the soil-specific calibration of the sensors, the first year of measurements, and additional measurements (soil temperature, phreatic groundwater levels and meteorological data) and
- 25 information (elevation, soil texture, land cover, and geohydrological model) available for performing scientific research. The data is available at http://dx.doi.org/10.4121/uuid:2411bbb8-2161-4f31-985f-7b65b8448bc9.

#### **1** Introduction

global to regional scales, the control of soil moisture on the exchanges of water and heat at the land surface plays an important role in the development of weather and climate systems (Global Climate Observing System, 2010; Seneviratne et 30 al., 2010). As a result, the Global Climate Observing System initiative (2010) has identified soil moisture as an Essential

Soil moisture is a hydrological state variable that affects various processes on the global, regional and local scales. On the

## Earth System Discussion Science Dpen Access

Climate Variable. However, soil moisture products from state-of-the-art Land Surface Models (LSMs) show large biases compared to in situ observations (Xia et al., 2014; Zheng et al., 2015) and large variation among different models (Dirmeyer et al., 2006; Xia et al., 2014). Xia et al. (2014) point out that particularly the soil moisture outcomes from LSMs still need improvement. In situ observations help to identify the shortcomings of LSMs and to improve model descriptions of related

processes.

Soil moisture also affects numerous hydrological and ecological processes that are essential for a wide spectrum of applications on the regional to local scales. Regional water management can benefit from timely and reliable soil moisture information: it improves quantifications of flood risks by its effect on rainfall estimations and streamflow predictions (Beck et al., 2009; Massari et al., 2014; Wanders et al., 2014) and negative anomalies to current plant water demands are an indicator for (the onset of) droughts (Carrão et al., 2016; Wilhite and Glantz, 1985). The agricultural sector depends on sufficient root zone soil water availability for crop growth, but also excess of soil water can lead to severe losses (Feddes et al., 1978). In addition, wet soil conditions are unfavourable for the trafficability of farmlands, which can jeopardize the timely execution of essential agricultural practices and cause structural damage of land (Batey, 2009; Hamza and Anderson, 2005). Lastly, soil moisture information is relevant to assess the effects of groundwater extractions (Ahmad et al., 2002; Zimmermann et al., 2017), drainage systems and irrigation systems.

Soil moisture conditions can be quantified using in situ instruments (Starr and Paltineanu, 2002; Vereecken et al., 2014), earth observations (Kornelsen and Coulibaly, 2013; Petropoulos et al., 2015), and land process models subject to atmospheric forcing terms (Albergel et al., 2012; De Lange et al., 2014; Srivastava et al., 2015; Vereecken et al., 2008). Of these methods, in situ measurements are the most accurate and can have a high temporal resolution when automated, but they lack the spatial support. In contrast, earth observations and process models provide areal estimates and enable the

- quantification of soil moisture across large spatial domains, but uncertainties regarding their soil moisture estimates are still subject of investigation. The success of soil moisture retrieval from earth observations by satellites strongly depends on the satellite's observation specifications, assumptions and parameter values adopted for the retrieval algorithms, and soil and vegetation cover conditions (e.g. Burgin et al., 2017; Chan et al., 2016; Das et al., 2014; Kerr et al., 2016; Pathe et al., 2009).
- In addition, microwave observations originate from only the upper portion of the soil, varying from surface to 0.001 m 0.125 m depth. This is controlled by the microwave wavelength, the sensor type (active or passive) and moisture conditions (Escorihuela et al., 2010; Nolan and Fatland, 2003; Rondinelli et al., 2015; Ulaby et al., 1996). Yet, several studies have reported that surface soil moisture can provide information about soil moisture at larger depths (Das and Mohanty, 2006; Ford et al., 2014; Vereecken et al., 2008). To relate surface soil moisture to soil moisture at larger depths, correct
- specification of hydraulic parameters and modelling of the hydrological processes are required (Chen et al., 2011; Das and Mohanty, 2006; Vereecken et al., 2008). Regarding land process models, the implemented model physics, model structure, the quality of parameterizations and the imposed initial and boundary conditions (including atmospheric forcing terms) determine the reliability of model results (Xia et al., 2014). Combining observations of earth variables with process models

# Science Science Science Science

by data assimilation techniques is interesting in estimating initial model states, model state updating and parameter calibration, thereby improving the accuracy (Houser et al., 2012; Reichle, 2008; Vereecken et al., 2008).

In situ soil moisture measurements provide a reference for validating earth observation retrievals and process models. The combination of in situ measurements at various depths, earth observation products and land process models is

- 5 essential to obtaining reliable soil moisture information at the temporal, horizontal and vertical resolutions required for the above-mentioned applications. Several regional scale soil moisture monitoring networks have been established to fulfil (part of) this aim. The International Soil Moisture Network (Dorigo et al., 2011) and the SMAP Cal/Val Partner Sites (Colliander et al., 2017) are two initiatives that bring together the data collected by a number of networks. The Natural Resources Conservation Service Soil Climate Analysis Network, consisting of 218 stations in agricultural areas across the United
- States of America, is operationally used for monitoring drought development, developing mitigation policies, predicting the long-term sustainability of cropping systems and watershed health, predicting regional shifts in irrigation water requirements and predicting changes in runoff (U.S. Department of Agriculture, 2016). Examples of regional scale networks in a temperate climate are the Little Washita (Cosh et al., 2006) and Little River (Bosch et al., 2007) networks in North America, REMEDHUS in Spain (Martínez-Fernández and Ceballos, 2005), Twente in the Netherlands (Dente et al., 2011, 2012; Van
- 15 der Velde et al., 2014), HOBE's network in Denmark (Bircher et al., 2012), SMOSMANIA in France (Albergel et al., 2008; Calvet et al., 2007) and TERENO in Germany (Zacharias et al., 2011) in Europe, and Kyeamba (Smith et al., 2012) in Australia.

This paper presents the soil moisture and temperature profile monitoring network in the Raam Catchment, in the southeast of the Netherlands, established in April 2016. For Dutch standards and in comparison to the only existing longterm soil moisture monitoring network in the Netherlands, in the Twente region (Dente et al., 2011, 2012; Van der Velde et al., 2014), the Raam Catchment faces substantial water shortage during summers. Extreme precipitation events cause an excess of water and inundation. The region contains intensive agriculture that is challenged by these extreme situations: agricultural yield largely depends on the applied water management. The Raam soil moisture monitoring network is

established jointly with the regional water management authority, Waterschap Aa en Maas. With the network, we aim to

- 25 collect data for calibration and validation of earth observation soil moisture products, assessment of land process model performance and understanding of processes affected by soil moisture (e.g. field trafficability, crop water availability). In addition, cooperation with the regional water management authority enables exploration of the potential of soil moisture information for optimizing operational regional water management. In this paper, we describe the characteristics of the Raam area (Sect. 2), the network design and instrumentation (Sect. 3), the sensor propagation distance (Sect. 4.1), the sensor
- 30 calibration results (Sect. 4.2), verification of the first year of measurements (Sect. 4.3), and data availability and additional data available for scientific research (Sect. 5).

#### 2 Study area

The Raam River is a tributary of the Meuse River with a catchment area of 223 km<sup>2</sup> that is situated in the southeast of the Netherlands; see Fig. 1a. The catchment has a temperate oceanic climate. On average, the coldest month is January (3.3 °C) and the warmest month is July (18.3 °C), based on measurements at Volkel weather station over the period 2000-2016 (Royal Netherlands Meteorological Institute (KNMI), 2017). Precipitation statistics are listed in Table 1. Precipitation in every month is roughly the same. Figure 2 shows the cumulative precipitation deficit for the hydrological year 2016 and the average over the period 2000-2015. The cumulative precipitation deficit is calculated by subtracting daily reference evapotranspiration rates from daily precipitation rates measured at the Volkel weather station and summing the daily deficits. A number of heavy precipitation events characterized May 2016 to August 2016, which caused the 2016 summer in the Raam to be wetter than normal. In dry years, the cumulative precipitation deficit can reach up to 100 mm in summer. During

- 10 Raam to be wetter than normal. In dry years, the cumulative precipitation deficit can reach up to 100 mm in summer. During these dry periods, farmers irrigate from deep groundwater reservoirs. The regional water management authority operates a system of weirs and pumping stations to minimize situations of excess water and droughts. In addition, the regional water management authority continuously discharges surface water into the southern part of the catchment to increase groundwater recharge. The average discharge into the catchment for the summer of 2016 was 900 m<sup>3</sup> h<sup>-1</sup>.
- The subsurface of the Raam Catchment consists of unconsolidated Pleistocene sandy and fluvial gravel sediments in two river terraces. The higher terrace slopes gently from 21 m a.s.l. to 17 m a.s.l. and the lower terrace slopes from 13 m a.s.l. to 8 m a.s.l., with the terrace edge lying in northwest-southeast direction (Fig. 1b). Remnants of peat and fine sands deposited by aeolian processes are found on the higher terrace. In parts of the study area, anthropogenic activities – the continuous addition of straw-mixed cattle droppings – have elevated fields, resulting in an approximately 1 m thick layer of
- 20 brown earth with high organic matter contents, called plaggen soils (Blume and Leinweber, 2004). Soils in the catchment are mostly sandy, with loam contents varying from 0% to about 20% (Wösten et al., 2013). In the eastern part, clayey soils are present (Fig. 1c). The main land covers are grassland (30%) and corn fields (20%), another 14% is used for other crops, built-up and paved areas occupy 14%, forests cover about 10% and open water covers 3%.
- Several northwest-southeast orientated dip-slip faults are present in the subsurface, as shown in Fig. 3. Movements along these faults have caused the formation of sharp lateral transitions between highly permeable and impermeable layers, as shown in Fig. 4. On the eastern part of the higher terrace (D-E in Fig. 3 and Fig. 4) this has resulted in the existence of a phreatic aquifer of only 10 m thick, whereas for the rest of the study area the phreatic aquifer is generally around 25 m to 50 m thick. The sharp transition in aquifer thickness leads to obstruction of the northeast-directed groundwater flow and high groundwater levels on the western part of the higher terrace (C-D), as shown in Fig. 3.

### Barth System Discussion Science Scienc

#### 3 Network design

#### **3.1 Station locations**

In April 2016, 15 stations were installed in the Raam Catchment. Figure 1 shows the locations of the stations. The locations for the stations were selected to capture the range of physical characteristics influencing the areas' hydrological dynamics.
The physical characteristics considered are soil texture (Sect. 3.1.1), land cover (Sect. 3.1.2) and elevation (Sect. 3.1.3). With 15 stations distributed over a 223 km<sup>2</sup> river catchment, the network's density translates to a coverage of 15 km<sup>2</sup> per station. Station 1 to station 5 are located in a closed sub-catchment of the Raam Catchment, called De Hoge Raam ('The High Raam'). The number of stations and the density of the Raam network are comparable to other soil moisture monitoring networks that are comparable in areal extent, such as the Little Washita network (20 stations, 30 km<sup>2</sup> average spacing), the Fort Cobb network (15 stations, 23 km<sup>2</sup> average spacing), the Reynolds Creek network (15 stations, 16 km<sup>2</sup> average

- spacing), the Little River network (33 stations, 10 km<sup>2</sup> average spacing) in the USA, and the Kyeamba network (14 stations, 43 km<sup>2</sup> average spacing) and the Adelong Creek network (5 stations, 29 km<sup>2</sup> average spacing), (Crow et al., 2012). Crow et al. (2012) state that these regional-scale networks facilitate validation over a range of land covers and at a scale which is finer than the footprints typical for operational coarse-scale soil moisture products from earth observations (ASCAT,
- 15 AMSR-2, SMOS and SMAP). Basin-scale aggregates are expected to have Root Mean Square Error ( $E_{RMS}$ ) values of 0.01 m<sup>3</sup> m<sup>-3</sup> (Crow et al., 2012), which is small compared to the  $E_{RMS}$  goal of 0.04 m<sup>3</sup> m<sup>-3</sup> defined for the SMOS mission (Kerr et al., 2010) and SMAP mission (Chan et al., 2016).

#### 3.1.1 Soil texture

The Raam Catchment mainly holds sandy soils. Therefore, 13 stations were positioned in coarse sandy soils. Two stations (station 6 and 7) were positioned in clayey sands and loamy sands respectively, at the northeastern part of the study area. Table 2 lists the soil type descriptions adopted from BOFEK2012. BOFEK2012 provides the soil physical characteristics (e.g. soil texture, water retention curve and hydraulic conductivity curve) for the soil units in the Netherlands, based on the Dutch class pedotransfer function known as the Staring series (Wösten et al., 2001, 2013). Table 2 also lists the corresponding World Reference Base soil order (Hartemink and De Bakker, 2006).

25 Complementary to the available soil texture information, we performed particle size analyses in a laboratory, following the pipette method described by Van Reeuwijk (2002), on samples representing the upper 40 cm of the soil profile at each soil moisture station. Organic matter content was determined by the Loss of Ignition (LOI) method (Davies, 1974; Hoogsteen et al., 2015) at 500 °C. The results reveal very high sand contents for most stations, and as expected, station 6 and 7 have higher volume fractions silt and clay. The results are consistent with the BOFEK2012 class descriptions.

#### 3.1.2 Land cover

Table 2 lists the land covers in 2016 of the fields adjacent to the soil moisture stations. Positioning of stations on agricultural areas was preferred over forest and natural areas. Microwave remote sensing instruments are typically unable to observe the soil under dense forest canopies, so measurements at agricultural areas are most valuable for validating soil moisture retrievals from earth observations. Furthermore, agricultural areas in particular are manageable regarding water related

retrievals from earth observations. Furthermore, agricultural areas in particular are manageable regarding water relate processes. Station 6 was positioned in natural grassland.

#### 3.1.3 Elevation

The stations were distributed in such a way that they cover the elevation gradient of the catchment. This will be valuable for observing the influence of groundwater level and water limited evapotranspiration conditions on soil moisture in the unsaturated zone.

#### 3.2 Instrumentation

Common instruments to measure volumetric soil moisture content are based on time-domain reflectometry (TDR) or capacitance techniques. Capacitance sensors are the most attractive choice for networks consisting of multiple soil moisture monitoring stations, because of their relatively low costs, ease of operation and applicability to a wide range of soil types

- (Bogena et al., 2007; Kizito et al., 2008; Vereecken et al., 2014). We deploy the Decagon 5TM capacitance sensor. The 5TM and other Decagon sensors using the same technique and frequency have been widely used for in situ soil moisture networks and have proved to fulfil the performance requirements (Bircher et al., 2012; Bogena et al., 2010; Dente et al., 2009, 2011; Kizito et al., 2008; Matula et al., 2016; Varble and Chávez, 2011; Vaz et al., 2013).
- 5TM sensors use an oscillator operating at 70 MHz to measure the capacitance of the soil, which is affected by the 20 soil's relative dielectric permittivity. The sensor prongs charge the surrounding soil, and the time needed to fully charge the soil defines the capacitance and consequently the relative dielectric permittivity of the soil. The relative dielectric permittivity of the soil varies as a function of the volumetric soil moisture content. Decagon Devices (2016) reports the following specifications for the 5TM: the resolution of the soil moisture measurements is 0.0008 m<sup>3</sup> m<sup>-3</sup> and the accuracy is  $\pm 0.03$  m<sup>3</sup> m<sup>-3</sup> for mineral soils by using the function established by Topp et al. (1980). A thermistor on the same probe measures soil temperature. The resolution of the temperature measurements is 0.1 °C and the accuracy is  $\pm 1$  °C.

The sensors are installed horizontally, with the prongs in vertical orientation to avoid ponding on the sensors due to water infiltration or condensation of vapour. Soil moisture and temperature are logged every 15 minutes with Decagon Em50 data loggers. At each location we installed 5TM sensors at depths of 5 cm, 10 cm, 20 cm, 40 cm and 80 cm; shown in Fig. 5.

Next at all soil moisture stations, phreatic groundwater level is monitored by Waterschap Aa en Maas at an hourly 30 time interval or by the Province of Noord-Brabant at a daily time interval with a MiniDiver DI501 (Schlumberger Water Services, 2014).

#### 3.3 Zone of influence on 5TM sensors

Although the sensing depth by microwaves is mostly from surface to the order of 0.01 m - 0.05 m depth (Escorihuela et al., 2010; Kornelsen and Coulibaly, 2013; Nolan and Fatland, 2003; Rondinelli et al., 2015; Ulaby et al., 1996), for practical reasons the shallowest in situ sensors are typically installed at 5 cm depth (Rondinelli et al., 2015; Shellito et al., 2016). In

- 5 air, 5TM sensors integrate a volume of 715 mL around the prongs, at maximum 6 cm from the centre of the sensor (Cobos, 2015). In soil, which has a higher dielectric permittivity, the outer edge will be closer to the sensor (Cobos, 2015). We performed an experiment to quantify the propagation distance of the waves from 5TM sensors in soil. Sakaki et al. (2008) and Cobos (2015) investigated the measurement volume in air and Vaz et al. (2013) in deionized water by moving the sensor towards/from a front of water and air respectively. We conducted the same kind of experiment in soil, by bringing a steel
- 10 knife, which has an extremely high dielectric permittivity, towards a 5TM sensor buried in the middle of a container with soil from station 1. The steel knife is brought towards the 5TM sensor from the direction at which in the field the soil surface would be. With this experiment we can leave the 5TM sensor in the same position to eliminate effects other than the steel knife. This procedure is performed five times for a range of soil moisture conditions.

#### **3.4 Calibration**

To convert sensor readings to volumetric soil moisture content we use the two-step calibration procedure (Bogena et al., 2007; Rosenbaum et al., 2010). The first step is the conversion of the sensor reading to relative dielectric permittivity. Kizito et al. (2008) concluded that there is no significant probe-to-probe variability among Decagon ECH<sub>2</sub>O-TE sensors, and Rosenbaum et al. (2010) found a  $E_{RMS}$  of approximately 0.01 m<sup>3</sup> m<sup>-3</sup> as a result of Decagon 5TE probe-to-probe variability. Decagon Devices calibrates each 5TM sensor, to account for probe-to-probe variability and to provide a linear relation between the sensor's response and the real part of the relative dielectric permittivity (Rosenbaum et al., 2010):

$$\varepsilon_a = \frac{5TM_{reading}}{50},\tag{1}$$

With  $5TM_{reading}$  [mV] being the raw output of the 5TM and  $\varepsilon_a$  [-] being the relative dielectric permittivity.

The second step is converting relative dielectric permittivity to volumetric soil moisture content. The relation between relative dielectric permittivity and soil moisture is affected by soil composition, bulk density, organic matter content
and soil salinity (Starr and Paltineanu, 2002). Relative dielectric permittivity can be converted to soil moisture using a general calibration function or using a soil-specific calibration function. By default the Decagon ECH2O Utility software applies the Topp function (Topp et al., 1980). However, Vaz et al. (2013) state that soil-specific calibration is often recommended to address the various soil property effects. According to Decagon Devices (2016) the accuracy can be improved from ± 0.03 m<sup>3</sup> m<sup>-3</sup> to ± 0.01 m<sup>3</sup> m<sup>-3</sup> – 0.02 m<sup>3</sup> m<sup>-3</sup> by using a soil-specific calibration. Indeed, several studies
concluded that soil-specific calibration can significantly improve the accuracy (e.g. Cosh et al., 2005; Dente et al., 2009, 2011; Varble and Chávez, 2011; Vaz et al., 2013).

Searth System Discussion Science Solutions Data

5

We developed soil-specific calibration functions for the main soil types present in the study area, by analysing samples taken from station 1, 7 and 10. The soil texture at these stations is considered representative for the soils at other stations; see Table 3. The measurements to establish the calibration function were collected following the procedure described by Starr and Paltineanu (2002), which employs pairs of gravimetrically determined volumetric soil moisture (GVSM) and sensor readings of relative dielectric permittivity. Under laboratory conditions, the GVSM and 5TM measurements were obtained in soil taken from station 1, 7 and 10, while gradually wetting the soil from air-dried conditions to saturated conditions by adding 75 ml to 100 ml of water. In every session typically 15 to 18 pairs of measurements were collected. The described procedure has been performed three times for each of the three soil samples.

The capability of the calibration functions to reproduce GVSM with 5TM measurements is evaluated with the coefficient of determination  $R^2$ ,  $E_{RMS}$ , and the bias, which we define as:

$$Bias = \frac{1}{n} \sum_{i=1}^{n} \theta_{5TM}(i) - \frac{1}{n} \sum_{i=1}^{n} \theta_{GVSM}(i),$$
(2)

With  $\theta_{5TM}$  [m<sup>3</sup> m<sup>-3</sup>] being the 5TM soil moisture reading converted by a calibration function, and  $\theta_{GVSM}$  [m<sup>3</sup> m<sup>-3</sup>] being the reference GVSM.

#### 4 Results and discussion

#### 15 4.1 Zone of influence

The results in Fig. 6 show that in soil the zone of influence ranges to 3 cm to 4 cm from the middle prong of the 5TM sensors. This is smaller than the propagation distance of 6 cm in air found by Cobos (2015) and larger than the propagation distance of 2.2 cm in deionized water found by Vaz et al. (2013). We conclude that at the shallowest installation depth of 5 cm open air does not affect the 5TM readings. The results indicate that soil moisture content does not affect the extent of the zone of influence.

#### 4.2 Calibration 5TM sensors

The results of the calibration procedure in Fig. 7 show that the 5TM readings and gravimetric measurements correlate well. The relations between the 5TM readings and GVSM can best be approximated by two-term power functions, because the relations are non-linear and, in contrast to polynomial functions, power functions keep increasing beyond the range of

25 GVSM obtained during the calibration procedure, which might occur in the field. The relation between relative dielectric permittivity sensor readings and volumetric soil moisture reads:

$$\theta_{cal} = a * \varepsilon_a{}^b + c, \tag{3}$$

With  $\theta_{cal}$  [m<sup>3</sup> m<sup>-3</sup>] being the calibrated volumetric soil moisture measurement,  $\varepsilon_a$  [-] being the measured relative dielectric permittivity, and *a*, *b* and *c* being calibration coefficients. The optimum calibration coefficients, listed in Table 3, are determined with the Metleb Curve Fitting Teelbox by Nep Linear Least Severes Fitting

determined with the Matlab Curve Fitting Toolbox by Non-Linear Least Squares Fitting.

Searth System Discussion Science Signate Discussions

Table 4 lists the associated error metrics. Lab calibration has reduced  $E_{RMS}$  from 0.03 m<sup>3</sup> m<sup>-3</sup> – 0.07 m<sup>3</sup> m<sup>-3</sup> to 0.02 m<sup>3</sup> m<sup>-3</sup> and has eliminated the bias between the 5TM readings and GVSM. The accuracy metrics using the Topp function are comparable or slightly worse than  $E_{RMS}$  values obtained by other studies using Decagon sensors, such as 0.06 m<sup>3</sup> m<sup>-3</sup> with EC-TM sensors by Dente et al. (2009), 0.05 m<sup>3</sup> m<sup>-3</sup> with EC-TM sensors by Dente et al. (2011), 0.04 m<sup>3</sup> m<sup>-3</sup> with 5TE sensors by Vaz et al. (2013), 0.02 m<sup>3</sup> m<sup>-3</sup> – 0.04 m<sup>3</sup> m<sup>-3</sup> with 5TE sensors by Varble and Chávez (2011), 0.02 m<sup>3</sup> m<sup>-3</sup> – 0.03 m<sup>3</sup> m<sup>-3</sup> with 5TE sensors by Bircher et al. (2012), and 0.03 m<sup>3</sup> m<sup>-3</sup> – 0.06 m<sup>3</sup> m<sup>-3</sup> with EC-5 sensors and 0.02 m<sup>3</sup> m<sup>-3</sup> – 0.04 m<sup>3</sup> m<sup>-3</sup> with 5TE sensors by Matula et al. (2016). The  $E_{RMS}$  values after the soil-specific calibration are comparable to values obtained by other studies that performed a soil-specific calibration (Dente et al., 2009, 2011; Kizito et al., 2008; Matula et al., 2016; Varble and Chávez, 2011; Vaz et al., 2013; Van der Velde, 2010).

Soil moisture measurements in the field (Fig. 8) exceed the maximum GVSM obtained at saturation conditions in the laboratory. Reasons may be higher organic matter contents, especially in the upper soil layers, roots and macropores present in the field.

#### 4.3 Data verification

#### 4.3.1 Data series completeness

The Raam network has generated data since April 2016. After 12 months of operations, 96% of the possible measurements are available. Data gaps are caused by probes not being properly connected for a time and malfunctioning of sensors and loggers (specified in a readme file attached to the measurement data).

#### 4.3.2 Data series analysis

We inspected the data from the 15 stations for possible errors. This includes an evaluation against the wilting point and 20 saturated soil moisture content for the soils in which the stations are placed. The wilting point and saturated soil moisture content are estimated using the Staring series (Wösten et al., 2001), which provides the Van Genuchten parameters for soil water retention and soil hydraulic conductivity. These parameters can be used to estimate soil moisture content for a specific pressure head using the Van Genuchten (1980) equation:

$$\theta(h) = \theta_r + \frac{\theta_s - \theta_r}{[1 + (\alpha|h|)^n]^{1-1/n}},\tag{4}$$

With *h* being the pressure head [cm of water],  $\theta(h)$  being the soil moisture content at pressure head *h* [m<sup>3</sup> m<sup>-3</sup>],  $\theta_r$  being the residual soil moisture content [m<sup>3</sup> m<sup>-3</sup>],  $\theta_s$  being the saturated soil moisture content [m<sup>3</sup> m<sup>-3</sup>],  $\alpha$  being a scale parameter inversely proportional to the air entry value [cm<sup>-1</sup>] and *n* being a parameter related to the pore size distribution [-]. BOFEK2012 provides the Staring series at the station locations (Wösten et al., 2013).

The boxplots in Fig. 8 show the range of the station measurements for each depth. The wilting point and saturated 30 soil moisture content are represented by red lines. Generally, the station measurements are within the range as expected based on BOFEK2012. The measurements of station 1, 8 and 13 slightly exceed the saturated soil moisture content, and

# Science Science Science Science

station 12 and 15 exceed the saturated soil moisture content to a larger extent. Furthermore, the measurements at 80 cm depth at station 1, 4, 6, 8 and 12 exceed the saturated soil moisture content for a long period. This may be explained by local soil variability that is not captured by BOFEK2012 and macroporosity that is not considered by BOFEK2012. As BOFEK2012 only considers soil matrix porosity, deviations may occur when additional cracks, biopores or other macropores exist.

Figure 9 shows a time series plot of soil moisture measurements at station 1 for all measured depths, along with daily precipitation data from the Volkel weather station. The soil moisture shows a clear response to the precipitation events. The soil moisture content at the upper layers shows larger dynamics than the soil moisture content at deeper layers. The soil moisture content at 80 cm is very stable, because it is controlled by the high phreatic groundwater level (GHG is 0.58 m

below surface at the location of station 1, Fig. 3).

Figure 10a shows the soil moisture content at 5 cm, 10 cm, 20 cm, 40 cm and 80 cm depth averaged over time and over all stations. The average soil moisture content increases with depth from 0.23 m<sup>3</sup> m<sup>-3</sup> at 5 cm to 0.30 m<sup>3</sup> m<sup>-3</sup> at 80 cm. Indeed, one can expect the top soil to be drier than the deeper parts due to infiltration and evapotranspiration. Figure 10b shows the relative standard deviation, which is defined as the ratio of the standard deviation to the average soil moisture

- content, for each depth averaged over time and over all stations. A higher relative standard deviation indicates larger 15 variability of soil moisture. Figure 10b indicates decreasing soil moisture variability with increasing depth, which was also visible for station 1 in Fig. 9. The upper layers are controlled by precipitation and evapotranspiration, which are relatively variable in time. The deeper layers are controlled by the generally high phreatic groundwater levels (Fig. 3), which provide a continuous source of water by capillary rise.
- The observed dynamics of soil moisture at catchment scale are as expected. However, local differences in surface elevation, soil composition and land cover play an important role in local scale variation. Over time, changes in land cover and macroporosity, and temperature effects (Sect. 4.3.3) introduce uncertainties.

#### 4.3.3 Effect of temperature

- The soil moisture series at 5 cm, 10 cm and 20 cm depth shows diurnal variations at all stations. There might be a soil hydrological cause, i.e. by dew or adsorption of water vapour by the soil, which causes an increase in soil moisture during 25 the night and morning (Agam and Berliner, 2006; Kosmas et al., 1998). Alternatively, the soil moisture sensors readings might be sensitive to temperature. A number of studies (Bogena et al., 2007; Kizito et al., 2008; Rosenbaum et al., 2011; Verhoef et al., 2006) found that the dielectric permittivity readings of soil moisture sensors are affected by soil temperature, varying from -0.002 m<sup>3</sup> m<sup>-3</sup> °C<sup>-1</sup> to 0.004 m<sup>3</sup> m<sup>-3</sup> °C<sup>-1</sup>. Figure 11 shows the largest soil moisture to temperature sensitivities
- measured at 5 cm depth between 8:00 at day 1 and 8:00 at day 2, under the following conditions: no precipitation on the day 30 itself and the preceding two days, a temperature difference at start time and end time of maximum 1.0 °C and a soil moisture difference at start time and end time of maximum 0.005 m<sup>3</sup> m<sup>-3</sup>. To calculate the sensitivity of soil moisture to temperature, the soil moisture series is linearly detrended first, assuming constant drainage and evaporation over the period of

5

15

investigation (Cobos and Campbell, 2007). At station 7 in wet conditions (Fig. 11a), the upward and downward slope of soil moisture have a time lag with respect to temperature. This suggests a soil hydrological process causes the diurnal variation of soil moisture, such as the addition of water by dew. At station 13 in dry conditions (Fig. 11b), there is probably a direct effect of temperature on the soil moisture signal. Over all stations and all diurnal cycles matching to the conditions introduced before, the average absolute sensitivity of soil moisture to temperature is 0.0006 m<sup>3</sup> m<sup>-3</sup> °C<sup>-1</sup>. The difference between the minimum and maximum daily average soil temperature at 5 cm over the measurement period 5 April 2016 to 4 April 2017 is 19 °C to 28 °C, translating to an effect of 0.011 m<sup>3</sup> m<sup>-3</sup> to 0.017 m<sup>3</sup> m<sup>-3</sup> on the soil moisture measurements by seasonal temperature variation. We consider this as a small effect compared to local variations and other measurement uncertainties, and since there might also be a soil hydrological cause, we do not correct for it.

#### 10 5 Data availability

The soil moisture are available 4TU.ResearchData and temperature data at the data centre at http://dx.doi.org/10.4121/uuid:2411bbb8-2161-4f31-985f-7b65b8448bc9. The data set currently covers the period between 5 April 2016 and 4 April 2017. Data collection will continue until at least October 2019. The data is stored in CSV files. A readme file describes the structure of the CSVs, contact information and metadata. Also included is a file containing information about additional data sets available for the Raam Catchment (elevation, soil physical, land cover, groundwater levels and meteorological data). Due acknowledgment in any publication or presentation arising from the use of these data is

#### **6** Conclusions

required.

The Raam soil moisture and temperature profile monitoring network contains 15 stations distributed over the 223 km<sup>2</sup> Raam Catchment. The stations consist of 5TM sensors installed at 5 cm, 10 cm, 20 cm, 40 cm and 80 cm depth. The measurements at 5 cm depth provide a reference for soil moisture retrievals from earth observations, and the measurements at deeper layers enable investigation of soil hydrological processes throughout the root zone. An experiment on the sensor's zone of influence proves that the sensor at 5 cm depth is not affected by open air. Soil-specific calibration functions for the 5TM sensors that have been developed under laboratory conditions lead to an accuracy of 0.02 m<sup>3</sup> m<sup>-3</sup>, compared to 0.03 m<sup>3</sup> m<sup>-3</sup> –

- 25 0.07 m<sup>3</sup> m<sup>-3</sup> with the Topp function. Verification of the first year of data shows that the station measurements are generally within the range as expected based on the classified soil units and associated soil physical characteristics from the soil map of the Netherlands (BOFEK2012). Exceedance of the upper limit occurs at stations 1, 4, 6, 8, 12, 13 and 15, which could be the effect of local soil variability not captured by BOFEK2012 and macroporosity not considered by BOFEK2012. The measurements show an expected soil moisture profile, with the average soil moisture increasing and soil moisture variability
- 30 decreasing with depth.

#### Acknowledgements

This work is part of the research programme OWAS1S (Optimizing Water Availability with Sentinel-1 Satellites) with project number 13871 which is partly financed by the Netherlands Organisation for Scientific Research (NWO). The regional water management authority Waterschap Aa en Maas contributed to the installation and maintenance of the network

and we thank Arjan Peters and Martijn van Helvert in particular. The authors also thank the field owners for their cooperation in granting access. Furthermore, the authors thank Caroline Lievens of University of Twente – Faculty of Geo-Information Science and Earth Observation for her help with the soil texture analysis in the laboratory.

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
