# Peer review of "Regional soil moisture monitoring network in the Raam catchment in the Netherlands"

_Earth System Science Data, 2017_

## Referee Comment (RC1) · Anonymous Referee #1 · 16 Aug 2017

pag 1 line 26: hard to find and download the data. Make the website easier to use or explain how to get them pag 2 line 13: trafficability of the lands is not so much scientifically interesting, as it is something more related to the farmers' necessities (to do it timely, and trying to avoid damages). more focus on e.g. groundwater/drainage/irrigation pag 2 line 23: the sentence is quite clear but could be divided in 2 sentences and explained a bit better. pag 2 line 25: it should be mentioned before that soil moisture retrivial is based on microwave pag 2 line 29-30: is it easy and effective the 'estimation' of deeper soil moisture? pag 4 line 6: maybe add a graph with the average monthly precipitation pag 5 line 3: in Figure1 stations 8 and 9 are outside the Raam catchment. Maybe provide a short explanation about it pag 5 lines 3-4: locations and stations are repetead very closely pag 5 line 14: maybe provide an example of the footprint, just to give an

idea pag 6 line 2: it is not clear if the stations are WITHIN the fields or adjacent. In the latter case is it important to know the land cover of the adjacent field? pag 6 line 26: if the prongs are installed with vertical orientation, the instrument is not measuring one depth (e.g 5 cm) but a range (e.g from 3 to 7 cm, etc.). This should be explained a bit more pag 8 line 22: the correlation (R2) could be added in the graphs of Fig. 7 pag 9 lines 3-6: a bit confusing pag 9 lines 10-12: not clear: the measurements in lab shouldn't be done on the samples collected in situ, i.e. with the same organic matter and macropores?? pag 9 par 4.3.2: maybe show/mention if the data exhibit a specific behaviour related to the area/conditions pag 10 lines 11-12: maybe it would be nice to see if there are any differences among the 3 stations used as reference (1, 7, 10) (different soil types)

page 30 table 2: column names could be more explicit (e.g soil classifications -columns 2 and 3, and ground data obtained from lab - columns 4 to 7)

---

## Referee Comment (RC2) · Anonymous Referee #2 · 18 Aug 2017

Regional soil moisture monitoring network in the Raam catchment in the Netherlands.

General comments: The study describes the implementation of a new in situ soil moisture monitoring network In the Raam catchment in the Netherlands. It is definitely relevant to the HESS journal. I think the paper is well written and well presented. I think the methodology is thorough and well explained, with a concise description of the calibration techniques employed. I have only minor comments regarding the validation. In particular, the data series analysis could be improved by demonstrating the influence of soil type and vegetation on the soil moisture measurements over the validation year.

Specific comments: Section 3.1: The stations are densely situated with stations located 15km apart on average and some stations just 2.5 km apart (e.g. 1 and 4). So you would presumably need a very high resolution land surface model or hydrological

model to resolve such small scale variability. Please give some examples, perhaps referencing studies for similar size networks.

Section 4.3.2: Information is missing regarding the influence of soil type and vegetation on the soil moisture measurements over the validation year. For example, for sandy soils you would expect to find a smaller dynamic range than clay soils. Was this evidenced in your results? It would be useful to see soil moisture time series plots for stations with different soil/vegetation types.

Section 4.3.3: Most soil moisture measuring devices malfunction when soils are frozen. This can lead to spurious low values (e.g. Hallikainen et al 1985). Did this affect any of the stations during the validation period and could this potentially be an issue? If so, would it be possible to plot the soil moisture for a station during frozen conditions?

Section 5: I also found the website a bit unintuitive. Please make it easier to find the data.

Section 6: In the conclusions section it might be good to add some information on future work that is expected to result from this study. What particular models/data assimilation systems might people be interested in using?

Figure 3: Perhaps use a different colour scale to show better the GHG variability Table 2: Could be refined a bit. References could be removed from the column headings and put in the caption instead.

References: Hallikainen, M. T., F. T. Ulaby, M. C. Dobson, M. A. El-Rayes, and L.-K. Wu, 1985: Microwave dielectric behavior of wet soil̆ĂŤ Part I: Empirical models and experimental observations. IEEE Trans. Geosci. Remote Sens., 23, 25–34, doi:10.1109/ TGRS.1985.289497.

---

## Author Comment (AC1) · 5 Oct 2017

**Response to Interactive comment Anonymous Referee #1     2017-10-05**

We thank the reviewer for the assessment. We appreciate the valuable suggestions provided to improve the manuscript. Below are our responses to the comments and points raised.
* * *
**(1) Comment:** *pag 1 line 26: hard to find and download the data. Make the website easier to use or explain how to get them.*

**Reply**: We agree with the comment. We will provide a direct link to the dataset: https://doi.org/10.4121/uuid:dc364e97-d44a-403f-82a7-121902deeb56 instead of https://doi.org/10.4121/uuid:2411bbb8-2161-4f31-985f-7b65b8448bc9. The first link leads to the dataset concerning the period 2016-04-05 to 2017-04-04. The second link leads to the so-called data collection. The dataset is part of this data collection. New data (e.g. the period 2017-04-05 to 2018-04-04) will be added to the data collection in due time. We propose the following changes to the manuscript.

In the abstract (page 1 line 25-26):

"The data  are available at https://doi.org/10.4121/uuid:dc364e97-d44a-403f-82a7-121902deeb56."

In Sect. 5 Data availability (page 11 line 11-13):

"The soil moisture and temperature data are available at the 4TU.ResearchData data centre at https://doi.org/10.4121/uuid:dc364e97-d44a-403f-82a7-121902deeb56. The data are found under the 'DATA' header. The data set currently covers the period between 5 April 2016 and 4 April 2017. New data will be added to the data collection at https://doi.org/10.4121/uuid:2411bbb8-2161-4f31-985f-7b65b8448bc9."
* * *
**(2) Comment:** *pag 2 line 13: trafficability of the lands is not so much scientifically interesting, as it is something more related to the farmers necessities (to do it timely, and trying to avoid damages). more focus on e.g. groundwater/drainage/irrigation*

**Reply:** We do not agree with the reviewer that trafficability is not scientifically interesting. Several studies show that soil trafficability is related to soil compaction, which is one of the leading causes of land degradation (Batey, 2009; Hamza and Anderson, 2005; Schwilch et al., 2016).
* * *
**(3) Comment:** *pag 2 line 23: the sentence is quite clear but could be divided in 2 sentences and explained a bit better.*

**Reply:** We want to make clear that there are three factors: 1. observation specifications of the satellite, 2. the assumptions and parameter values adopted for the retrieval algorithm, and 3. soil and vegetation cover conditions. We propose the following changes to improve readability of the sentence (page 2 line 22-24):

"The success of soil moisture retrieval from earth observations  depends on the specifications of the sensor, the assumptions and parameter values adopted for the retrieval algorithms, and the soil and vegetation cover conditions (e.g. Burgin et al., 2017; Chan et al., 2016; Das et al., 2014; Kerr et al., 2016; Pathe et al., 2009)."
* * *
**(4) Comment:** *pag 2 line 25: it should be mentioned before that soil moisture retrieval is based on microwave*

**Reply:** We agree with the reviewer that this is not explained clearly. In fact, observations in the optical spectrum (reflectance-based and thermal infrared-based methods), which are used for soil moisture estimation too, also originate from the upper portion of the soil (Petropoulos et al., 2015). In addition, it should be mentioned that earth observations are affected by the vegetation. Observation of vegetation characteristics by microwave sensors and optical sensors even have potential to provide indirect estimates of root zone soil moisture (Van Emmerik et al., 2015; Petropoulos et al., 2015; Steele-Dunne et al., 2012; Wang et al., 2010).

Besides, we noted an inconsistency in the manuscript: in Sect. 1 we mention a sensing depth of 0.001 m - 0.1  m (rough estimate) and in Sect. 3.3 we mention a sensing depth of 0.01 m - 0.05 m (supported by literature).

We propose to make the following changes in Sect. 1 (page 2 line 25-31):

" Microwave observations from X-band to L-band, which are most often used for the direct estimation of soil moisture by earth observations (Kornelsen and Coulibaly, 2013; Petropoulos et al., 2015), originate from the soil surface to 0.01 m - 0.05 m depth (Escorihuela et al., 2010; Kornelsen and Coulibaly, 2013; Nolan and Fatland, 2003; Rondinelli et al., 2015; Ulaby et al., 1996).  This is controlled by the microwave wavelength, the sensor type (active or passive) and the soil moisture conditions (Escorihuela et al., 2010; Nolan and Fatland, 2003; Rondinelli et al., 2015; Ulaby et al., 1996). Estimations of vegetation characteristics, by microwave and optical sensors, have potential to provide indirect estimates of root zone soil moisture (Van Emmerik et al., 2015; Petropoulos et al., 2015; Steele-Dunne et al., 2012; Wang et al., 2010). In addition, several studies have reported that surface soil moisture  may provide information about soil moisture at larger depths (Das and Mohanty, 2006; Ford et al., 2014; Vereecken et al., 2008)."

In Sect. 3.3 (page 7 line 2-4):

"For practical reasons the shallowest in situ sensors are typically installed at 5 cm depth (Rondinelli et al., 2015; Shellito et al., 2016)."
* * *
**(5) Comment:** *pag 2 line 29-30: is it easy and effective the 'estimation' of deeper soil moisture?*

**Reply:** We interpret this comment in the following way: currently we do not explain that the relation between surface soil moisture and soil moisture at deeper layers is not straightforward. We propose to add a sentence at page 2 line 29:

"In addition, several studies have reported that surface soil moisture  may provide information about soil moisture at larger depths (Das and Mohanty, 2006; Ford et al., 2014; Vereecken et al., 2008). However, the relation between surface soil moisture and soil moisture at deeper layers is complicated. To relate surface soil moisture to soil moisture at larger depths, correct specification of hydraulic parameters and modelling of the hydrological processes are required (Chen et al., 2011; Das and Mohanty, 2006; Vereecken et al., 2008)."
* * *
**(6) Comment:** *pag 4 line 6: maybe add a graph with the average monthly precipitation*

**Reply**: We agree that a figure with average monthly precipitation has added value. We will add a subplot with the average monthly precipitation to Fig. 2 and propose the following changes in Sect. 2 (page 4 line 5-7)

"Precipitation statistics are listed in Table 1. Figure 2a shows the average monthly precipitation measured at the Volkel weather station for the period 2000-2015 and for the hydrological year 2016.  Figure 2b shows the average cumulative precipitation deficit for the period 2000-2015 and for the hydrological year 2016

[Figure]

**Figure 2: (a) Average monthly precipitation for the period 2000-2015 and the monthly precipitation in the hydrological year 2016 measured at KNMI's Volkel weather station. (b) Daily and cumulative precipitation deficits for the period 2000-2015 and for the hydrological year 2016 , based on  precipitation measurements and reference evapotranspiration calculations at Volkel weather station."**
* * *
**(7) Comment:** *pag 5 line 3: in Figure1 stations 8 and 9 are outside the Raam catchment. Maybe provide a short explanation about it*

**Reply:** This is a good point. We looked at the catchment boundaries again. It appeared that the boundaries in Fig. 1 in the manuscript are administrative borders (municipalities), instead of hydrological boundaries. We propose to replace these boundaries with hydrological catchment boundaries based on a digital elevation model and artificial drainage areas. The sharp angular appearance of the newly defined boundaries at some points is because of artificial drainage areas, such as agricultural fields that have water drainage to a specific direction.

Besides, we would like to add the catchment boundaries of the sub-catchment Hoge Raam. We propose to replace Fig. 1:

Old figure:

[Figure]

**Figure 1: a. Location of the Raam study area (black box) in the Netherlands, b. Digital Elevation Model (Actueel Hoogtebestand Nederland, 2016), c. Major soil types classes (BOFEK2012, Wösten et al. (2013))**

New figure:

[Figure]

**Figure 1: a. Location of the Raam study area (black box) in the Netherlands, b. Digital Elevation Model (Actueel Hoogtebestand Nederland, 2016), c. Major soil types classes (BOFEK2012, Wösten et al. (2013))**

This means that stations 8, 9 and 11 are outside the Raam catchment. The reason for this is the availability of suitable fields and farmers that are granting access. All stations can be used as ground-truth data for the validation of earth observation retrievals, and for studying unsaturated zone processes and trafficability of lands. Stations 1 to 8, 10 and 12 to 15 can be used for catchment modelling. We propose the following changes in Sect. 3.1 (page 5 line 3-8, also see comment 8):

"In April 2016, 15 stations were installed in the Raam region (Fig. 1).  The locations  were selected to capture the range of physical characteristics influencing the areas' hydrological dynamics. The physical characteristics considered are soil texture (Sect. 3.1.1), land cover (Sect. 3.1.2) and elevation (Sect. 3.1.3). Stations 1 to 8, 10 and 12 to 15 are located within

the Raam Catchment. Stations 1 to 5 are located in a closed sub-catchment of the Raam Catchment, called De Hoge Raam ('The High Raam'). With 15 stations distributed over a 223 km² area, the network's density translates to a coverage of 15 km² per station. "

In Sect. 6 (page 11 line 30, also see comment 6 of Reviewer #2):

"The soil moisture measurement series of the Raam monitoring network provide a valuable data set for researching water management applications of soil moisture information, for validation of earth observation retrievals at coarse-scale and field scale, for studying processes in the unsaturated zone, and for validation of land process models. Stations 1 to 8, 10 and 12 to 15 can be used for modelling the catchment behaviour of the Raam Catchment."
* * *
**(8) Comment:** *pag 5 lines 3-4: locations and stations are repeated very closely*

**Reply:** We agree with this comment and propose the following changes (page 5 lines 3-4):

"In April 2016, 15 stations were installed in the Raam region (Fig. 1).  The locations  were selected to capture the range of physical characteristics influencing the areas' hydrological dynamics."
* * *
**(9) Comment:** *pag 5 line 14: maybe provide an example of the footprint, just to give an idea*

**Reply:** We agree with this suggestion and propose the following changes (page 5 line 12-15):

"Crow et al. (2012) state that these regional-scale networks provide information over a range of land covers and at a scale that allow the validation of  operational coarse-scale soil moisture products from earth observations  such as ASCAT at 25 km and 50 km (Wagner et al., 2013), AMSR-2 at 0.1° and 0.25° (Zhang et al., 2017), SMOS at 43 km (Kerr et al., 2016) and SMAP at 40 km resolution (Chan et al., 2016)."
* * *
**(10) Comment:** *pag 6 line 2: it is not clear if the stations are WITHIN the fields or adjacent. In the latter case is it important to know the land cover of the adjacent field?*

**Reply**: The stations are installed at a border of fields, so that they do not hinder farming practices. If the adjacent field is a grass field, the sensors typically are installed at a location with a similar (grass) cover. In case of crop fields, installation at a field with similar land cover is usually not possible and the sensors are installed at locations with either grass or bare cover. For readability reasons, we have split Table 2 in two separate tables, for more information we refer to comment 8 of Reviewer #2. The new Table 2 concerns soil type and the new Table 3 concerns land cover. We will add the land cover of the exact location of the station to Table 3 and we propose the following changes:

In Sect. 3.1.2 (page 6 line 2):

"The soil moisture stations were installed at the border of fields for practical reasons. Table 3 lists the land cover of the adjacent fields in 2016 as well as the land cover at the exact location of the soil moisture stations in 2016."

New Table 3:

**Table 3: Land cover near the soil moisture monitoring stations**

| Station | Land cover of adjacent field in 2016 | Land cover at location of station in 2016 |
|---|---|---|
| 1 | Grass | Grass |
| 2 | Sugar beets | Grass |
| 3 | Grass | Grass |
| 4 | Grass | Grass |
| 5 | Onions | Grass fallow |
| 6 | Natural grass | Grass |
| 7 | Corn & Cichorium | Grass fallow |
| 8 | Sugar beets | Grass |
| 9 | Sugar beets | Grass fallow |
| 10 | Grass | Grass |
| 11 | Corn & Grass | Grass |
| 12 | Grass | Grass |
| 13 | Corn | Grass |
| 14 | Grass | Grass |
| 15 | Grass | Grass |
* * *
**(11) Comment:** *pag 6 line 26: if the prongs are installed with vertical orientation, the instrument is not measuring one depth (e.g 5 cm) but a range (e.g from 3 to 7 cm, etc.). This should be explained a bit more*

**Reply:** This observation is correct. This is described in Sect. 3.3 and Sect. 4.1. To stress the consequences of this observation, we propose to add the following to Sect. 6 (page 11 line 20-22):

"The stations consist of 5TM sensors installed at 5 cm, 10 cm, 20 cm, 40 cm and 80 cm depth. The measurements at 5 cm depth provide a reference for the surface soil moisture retrievals from earth observations, and the measurements at deeper layers enable investigation of soil hydrological processes throughout the root zone. The experiment on the sensor's zone of influence shows that the sensor integrates a soil volume of 3 cm to 4 cm above and below the sensor's middle prong, so the installation depth of 5 cm is required to avoid effects of the open air. An experiment on the sensor's zone of influence proves that the sensor at 5 cm depth is not affected by open air."

Besides, we would like to point out that according to the results of Cobos (2015), the zone of influence parallel to the prongs ranges up to 6 cm and perpendicular to the prongs up to 5 cm. This means that also in horizontal direction the sensor measures a volume around the sensor. This is even advantageous, because it reduces the effect of small-scale heterogeneities in the soil (Cobos, 2015).
* * *
**(12) Comment:** *pag 8 line 22: the correlation (R2) could be added in the graphs of Fig. 7*

**Reply:** We agree that this adds relevant information and have included R² in Fig. 7:

[Figure]

**Figure 7: Decagon 5TM dielectric permittivity readings against GVSM, measured in the laboratory in soil from a selection of fields. The power fits are used for converting the relative dielectric permittivity measurements by 5TM sensors to volumetric soil moisture content.**
* * *
**(13) Comment:** *pag 9 lines 3-6: a bit confusing*

**Reply:** We agree that these lines are difficult to read and therefore propose the following table instead of the text (page 9 line 2-9):

"The accuracy metrics using the Topp function are comparable or slightly worse than $E_{RMS}$ values obtained by other studies using Decagon sensors~~, , such as 0.06 m³·m⁻³ with EC-TM sensors by Dente et al. (2009), 0.05 m³·m⁻³ with EC-TM sensors by Dente et al. (2011), 0.04 m³·m⁻³ with 5TE sensors by Vaz et al. (2013), 0.02 m³·m⁻³ – 0.04 m³·m⁻³ with 5TE sensors by Varble and Chávez (2011), 0.02 m³·m⁻³ – 0.03 m³·m⁻³ with 5TE sensors by Bircher et al. (2012), and 0.03 m³·m⁻³ – 0.06 m³·m⁻³ with EC-5 sensors and 0.02 m³·m⁻³ – 0.04 m³·m⁻³ with 5TE sensors by Matula et al. (2016). The~~ and the $E_{RMS}$ values after the soil-specific

calibration are comparable to the values obtained by other studies that performed a soil-specific calibration (Table X).

**Table X: Error metrics between the GVSM and the readings by various Decagon sensors reported in former studies**

| Study | Study area and soil type | Sensor | $E_{rms}$ with Topp function [m$^3$ m$^{-3}$] | $E_{rms}$ with soil-specific calibration function [m$^3$ m$^{-3}$] |
|---|---|---|---|---|
| Bircher et al. (2012) | Western Denmark, podzol sandy and loamy soils | 5TE | Agricultural land: 0.030 Forest: 0.026 Heath: 0.022 | Not reported |
| Dente et al. (2009) Su et al. (2011) | Maqu, Tibetan Plateau, organic and silt loam soils | EC-TM | 0.06 | 0.02 |
| Dente et al. (2011) Dente et al. (2012) | Twente, the Netherlands, sand and loamy sand | EC-TM | 0.054 | 0.023 |
| Kizito et al. (2008) | Oso Flaco USA, sand Columbia, USA, silt loam | TE | Not reported | Combined: 0.026 Sand: 0.015 Silt loam: 0.018 |
| Matula et al. (2016) | Prague, Czech Republic, Haplic chernozem substrate loess | EC-5 TE | EC-5: 0.031 TE: 0.029 | EC-5: 0.018 TE: 0.023 |
| Van der Velde et al. (2012) | Naqu, Tibetan Plateau | EC-10 | Not reported | 0.029 |
| Vaz et al. (2013) | Arizona, USA, sandy to clayey soils | 5TE | 0.040 | 0.026 |
| Varble and Chávez (2011) | Colorado, USA | 5TE | Sandy clay loam: 0.022 Loamy sand: 0.025 Clay loam: 0.038 | Sandy clay loam: 0.021 Loamy sand: 0.007 Clay loam: 0.028 |

"
* * *
**(14) Comment:** *pag 9 lines 10-12: not clear: the measurements in lab shouldn0 t be done on the samples collected in situ, i.e. with the same organic matter and macropores??*

**Reply:** We performed the calibration procedure as recommended by Decagon. The calibration function describes the relation between the dielectric permittivity as measured by the sensor and soil moisture. This relation depends on soil texture, bulk density and organic matter. By taking soil from the field into the lab we have soil with the same soil texture and organic matter content. Field bulk density is approximated in the lab. As we used disturbed soil samples from the field to perform the calibration procedure, the exact conditions at the location of the sensors regarding macropores and roots can never be reproduced during the calibration. This is allowed because macropores do affect the soil moisture, but they do not affect the relation between the dielectric permittivity and soil moisture.

Nevertheless, macropores and roots may have an effect on the measurements and introduce local uncertainties. If the sensor is installed at a location with many or large macropores, this increases the saturated soil moisture content. Also the presence of roots (contain much water) increases soil moisture in the field. Therefore, higher soil moisture contents may be recorded in the field than in the lab.

We propose the following changes in the manuscript:

In Sect. 3.4 (page 8 line 3-7):

"The measurements to establish the calibration function were collected following the procedure described by Starr and Paltineanu (2002), as recommended by Decagon , (Cobos and Chambers, 2010). which The procedure employs pairs of gravimetrically determined volumetric soil moisture (GVSM) and sensor readings of relative dielectric permittivity. Under laboratory conditions, the GVSM and 5TM measurements were obtained in disturbed soil samples taken from stations 1, 7 and 10, while gradually wetting the soil from air-dried conditions to saturated conditions by adding 75 ml to 100 ml of water."

In Sect. 4.2 (page 9 line 10-12):

"Soil moisture measurements recorded in the field (Fig. 8) exceed the maximum GVSM obtained at saturation conditions in the laboratory. Reasons may be higher organic matter contents, especially in the upper soil layers, roots and macropores present in the field. Reasons may be the presence of roots and macropores in the field, which can never be reproduced with the disturbed soil samples used for the calibration procedure. In the field, macropores may be present, which increase the saturated soil moisture content. Also the presence of large roots increase recorded water contents."
* * *
**(15) Comment:** *pag 9 par 4.3.2: maybe show/mention if the data exhibit a specific behaviour related to the area/conditions*

**Reply**: We agree with this comment. Based on comment 3 of Reviewer #2, we plotted soil moisture measurements averaged for different soil types, groundwater depths and vegetation characteristics. We propose to add the following text and figure to the manuscript:

In Sect. 4.3.2 (page 10 line 23):

"We explored the influence of various factors on soil moisture dynamics. Figure X shows the average of soil moisture measurements at 20 cm at stations with a specific characteristic. Fig. Xa shows the average soil moisture content for stations in sandy soils and for stations in loamy/clayey soils. We expect sandy soils to have lower and more dynamic soil moisture contents than loamy/clayey soils, which Fig. Xa confirms. Fig. Xb shows that locations with deep groundwater levels (> 1 m) generally are drier than locations with shallow groundwater levels (< 1 m). The situation of shallow groundwater levels applies to the stations 1, 6, 8, 11, 12, 13 and 15, based on groundwater level measurements by the regional water authority Aa en Maas. Fig. Xc shows the variation in soil moisture content due to different vegetation types. In general, the soil moisture content is larger in corn fields in comparison with the other vegetation types. Also, grasslands tend to be wetter than fields with sugar beets and onions in the winter period of 2016/2017."

[Figure]

**Figure X: Influence of (a) soil type (Table 2), (b) groundwater depth (based on groundwater level measurements by the regional water authority Aa en Maas) and, (c) vegetation type (Table 3) on soil moisture dynamics at 20 cm depth."**
* * *
**(16) Comment:** *pag 10 lines 11-12: maybe it would be nice to see if there are any differences among the 3 stations used as reference (1, 7, 10) (different soil types)*

**Reply:** We agree that additional analysis to the soil moisture dynamics would be useful. A comparison is made in the figure as mentioned in the response to comment 15.
* * *
**(17) Comment:** *page 30 table 2: column names could be more explicit (e.g soil classifications -columns 2 and 3, and ground data obtained from lab - columns 4 to 7)*

**Reply:** We agree with this comment. We clarified the column names and will adapt the table as suggested in our response to comment 8 of Reviewer #2.

---

## Author Comment (AC2) · 5 Oct 2017

**Response to Interactive comment Anonymous Referee #2 2017-10-05**

Regional soil moisture monitoring network in the Raam catchment in the Netherlands.

**General comments:**

(1) Comment: The study describes the implementation of a new in situ soil moisture monitoring network in the Raam catchment in the Netherlands. It is definitely relevant to the HESS journal. I think the paper is well written and well presented. I think the methodology is thorough and well explained, with a concise description of the calibration techniques employed. I have only minor comments regarding the validation. In particular, the data series analysis could be improved by demonstrating the influence of soil type and vegetation on the soil moisture measurements over the validation year.

**Reply:** We thank the reviewer for the assessment. We appreciate the valuable suggestions provided to improve the manuscript. Below are our responses to the comments and points raised.

Regarding the comment on the data series analysis we refer to our response to comment 3.

**Specific comments:**

(2) Comment: Section 3.1: The stations are densely situated with stations located 15km apart on average and some stations just 2.5 km apart (e.g. 1 and 4). So you would presumably need a very high resolution land surface model or hydrological model to resolve such small scale variability. Please give some examples, perhaps referencing studies for similar size networks.

**Reply**: We refer to Sect. 3.1, where six networks, which are comparable in density, are listed. A soil moisture network of such density is advantageous for various reasons:

- The measurements can be used for validation of coarse-scale soil moisture products such as SMOS and SMAP. Soil moisture can exhibit a large spatial variability; the sub-footprint spatial standard deviation of point-scale soil moisture measurements often exceeds the  $E_{RMS}$  accuracy goal for SMOS and SMAP ( $E_{RMS} = 0.04 \text{ m}^3 \text{ m}^{-3}$ ) (Crow et al., 2012; Famiglietti et al., 2008). Multiple soil moisture measurements within the footprint of a coarse-scale soil moisture product reduces the measurement uncertainty of a footprint-scale soil moisture reference. We explain this in Sect. 3.1 of the manuscript (also see the proposed change at comment 9 of Reviewer #1).
- In hydrological research there is a trend towards hyperresolution land surface modelling (Beven et al., 2015; Wood et al., 2011). Wood et al. (2011) propose to have land surface models at continental scales that have a grid resolution of 100 m by 100 m. Another example is the Netherlands Hydrological Model (NHI) that is currently operating at a spatial resolution of 250 m by 250 m (De Lange et al., 2014).
- Stations 1 to 5 are located in the sub-catchment Hoge Raam ('High Raam') of the Raam catchment, which is relevant for hydrological catchment studies. We refer to comment 7 of Reviewer #1 for a more in-depth explanation.

(3) Comment: Section 4.3.2: Information is missing regarding the influence of soil type and vegetation on the soil moisture measurements over the validation year. For example, for sandy soils you would expect to find a smaller dynamic range than clay soils. Was this evidenced in your results? It would be useful to see soil moisture time series plots for stations with different soil/vegetation types.

**Reply**: We agree with this comment. We have plotted soil moisture measurements averaged for different soil types, groundwater depths and vegetation characteristics. We propose to add the following text and figure to the manuscript:

In Sect. 4.3.2 (page 10 line 23):

"We explored the influence of various factors on soil moisture dynamics. Figure X shows the average of soil moisture measurements at 20 cm at stations with a specific characteristic. Fig. Xa shows the average soil moisture content for stations in sandy soils and for stations in loamy/clayey soils. We expect sandy soils to have lower and more dynamic soil moisture contents than loamy/clayey soils, which Fig. Xa confirms. Fig. Xb shows that locations with deep groundwater levels (> 1 m) generally are drier than locations with shallow groundwater levels (

Figure X: Influence of (a) soil type (Table 2), (b) groundwater depth (based on groundwater level measurements by the regional water authority Aa en Maas) and, (c) vegetation type (Table 3) on soil moisture dynamics at 20 cm depth."

(4) **Comment:** Section 4.3.3: Most soil moisture measuring devices malfunction when soils are frozen. This can lead to spurious low values (e.g. Hallikainen et al 1985). Did this affect any of the stations during the validation period and could this potentially be an issue? If so, would it be possible to plot the soil moisture for a station during frozen conditions?

**Reply:** It is correct that the soil moisture sensors do not record reliable soil moisture values when soils are frozen. However, we would like to note that the sensors do not really malfunction. When soils are (partly) frozen, the free water content decreases and this affects the bulk dielectric permittivity (which is what soil moisture sensors actually measure). The figure below shows an example of this phenomenon. During periods when the temperature, which is measured by the same device, is close to 0 °C, the soil moisture content as measured by the sensor decreases sharply. The exact temperature below which soil moisture measurements are affected depends on the soil moisture, soil texture, and the temperature profile (Watanabe and Flury, 2008). Therefore, it is tricky to give a threshold temperature below which the soil moisture measurements are affected.